# Attitudes towards Participation in a Passive Data Collection Experiment

**DOI:** 10.3390/s21186085

**Published:** 2021-09-10

**Authors:** Bence Ságvári, Attila Gulyás, Júlia Koltai

**Affiliations:** 1Computational Social Science—Research Center for Educational and Network Studies (CSS–RECENS), Centre for Social Sciences, Tóth Kálmán Utca 4, 1097 Budapest, Hungary; Attila.gulyas@tk.hu (A.G.); Julia.koltai@tk.hu (J.K.); 2Institute of Communication and Sociology, Corvinus University, Fővám tér 8, 1093 Budapest, Hungary; 3Department of Network and Data Science, Central European University, Quellenstraße 51, 1100 Vienna, Austria; 4Faculty of Social Sciences, Eötvös Loránd University of Sciences, Pázmány Péter Sétány 1/A, 1117 Budapest, Hungary

**Keywords:** data fusion, surveys, informed consent

## Abstract

In this paper, we present the results of an exploratory study conducted in Hungary using a factorial design-based online survey to explore the willingness to participate in a future research project based on active and passive data collection via smartphones. Recently, the improvement of smart devices has enabled the collection of behavioural data on a previously unimaginable scale. However, the willingness to share this data is a key issue for the social sciences and often proves to be the biggest obstacle to conducting research. In this paper we use vignettes to test different (hypothetical) study settings that involve sensor data collection but differ in the organizer of the research, the purpose of the study and the type of collected data, the duration of data sharing, the number of incentives and the ability to suspend and review the collection of data. Besides the demographic profile of respondents, we also include behavioural and attitudinal variables to the models. Our results show that the content and context of the data collection significantly changes people’s willingness to participate, however their basic demographic characteristics (apart from age) and general level of trust seem to have no significant effect. This study is a first step in a larger project that involves the development of a complex smartphone-based research tool for hybrid (active and passive) data collection. The results presented in this paper help improve our experimental design to encourage participation by minimizing data sharing concerns and maximizing user participation and motivation.

## 1. Introduction

Smartphone technologies combined with the improvement of cloud-based research architecture offers great opportunities in social sciences. The most common methodology in the social sciences is still the use of surveys and other approaches that require the active participation of research subjects. However, there are some areas that are best researched not through surveys, but rather by observing individuals’ behaviour in a continuous social experiment. Mobile technologies make it possible to observe behaviour on a new level by using raw data of various kinds collected by our most common everyday companion: our smartphone. Moreover, since smartphones shape our daily lives thanks to various actions available through countless apps, it is logical to consider them as a platform for actual research. 

There have been numerous research projects that have relied on collecting participants’ mobile sensor and app usage data, but the biggest concern has been the willingness to share this data. Privacy and trust concerns both contribute to people’s unwillingness to provide access to their personal data, and uncovering these attitudes is a critical step for any successful experimental design. 

In this paper, we present the results of our pre-experimental survey to uncover prospective participants’ attitudes toward sharing their mobile sensor and app usage data. This experiment is part of a larger research and software development project aimed at creating a modular active and passive data collection tool for smartphones that could be used in social and health research. 

For this study we used data from an online survey representative of Internet users in Hungary. The aim of the survey was to analyse respondents’ attitudes (and not actual behaviour) towards using a hypothetical research app that performs active and passive data collection. 

The following section provides further details on the background of active/passive data collection and an outlook on results from other studies. We then discuss the details of the online panel used in our study, the survey design and the models used in the analysis. After presenting our results, we conclude by mentioning some open questions and limitations that can be addressed in further steps of this study.

## 2. Background

### 2.1. Surveys, Active and Passive Data Collection

In recent decades survey methods have been the main research tools in the social sciences. Technological advances have not changed that, but rather expanded it. Traditional paper-and-pencil interviews (PAPI) and surveys quickly adopted new technologies: Interviews were conducted over telephones (regular surveys) and as computers became mainstream, computer aided survey methods emerged. 

This development took another leap when smartphone applications emerged along with cloud-based services and smartphones suddenly became a viable platform for collecting survey data [1,2,3,4]. Although self-reported surveys generally suffer from bias for a variety of reasons [5,6,7,8], conducting surveys with smartphones is a very cost-effective method of data collection that also opens up opportunities to collect other non-survey types of data. Such data includes location information, application usage, media consumption etc. all of which provide better insight into the behaviour and social connections of individuals [9,10,11,12,13]. More importantly, since it is behavioural data, it is much less prone to bias, unlike ordinary surveys.

The collection of data is divided into two main categories depending on the subscriber’s interaction with their smartphone: active and passive data collection.

Active data collection means that an action by the participant is required to generate the collected information, and the participant is prompted by the research application to provide this information. This means that the participant triggers phone features (taking photos, recording other types of data, actively sending a location tag) while also giving consent for this data to be sent to the researching institution. Submitting surveys or survey-like inputs (e.g., gathering attitudes or moods) [14,15,16]) can also be considered a form of active data collection. 

Passive data collection, on the other hand, means that sensor data from the smartphone is collected and sent periodically without the participant knowing that data was collected at any given time. There are various sensors that can be used in a smartphone: multiple location-based sensors (GPS, gyroscopes), accelerometers, audio sensors, Bluetooth radios, Wi-Fi antennas, and with the advancement of technology, many other sensors–such as pulse or blood pressure sensors. In the field of healthcare, such passive data collection is becoming the main solution for health monitoring in the elderly or in other special scenarios [17,18,19].

Obviously, such data collection approaches can be combined to provide instant data linkages [20], which can then be used to provide even richer information-e.g., pulse measurements while conducting surveys and answering questions can validate responses. 

In order to conduct such data collection in a legally and ethically acceptable manner, informed consent must be given by participants for every aspect of the data collection. With the inclusion of the GDPR, there are clear requirements for recording participants’ consents and handling their data, the key feature being that they can withdraw their consent at any time during their participation in an experiment.

For smartphone apps, the default requirement is that access to data and sensor information must be explicitly permitted by the user of the device. However, this consent does not apply to the sharing of data with third parties - in this case, the researching institution. The participants must give their explicit consent for their data to be collected and transferred from their device to a location unknown to them. Similarly, the researching institution must ensure proper handling of the data and is responsible to the participants for the security of their data. 

Several studies have found that people are generally reluctant to share their data when it comes to some form of passive data collection [21,22,23], mostly due to privacy concerns. However, people who frequently use their smartphones are less likely to have such concerns [23]. Over the past decade, the amount of data collected by various organizations has increased dramatically. This includes companies with whom users share their data with their consent [24], but they are probably unaware of the amount of data they are sharing and how it is exactly exploited for commercial purposes.

Several studies have found that people are much more likely to share data when they are actively engaged in the process (e.g., sending surveys, taking photos, etc.) than when they passively share sensor data [15,23]. This lower participation rate is influenced by numerous factors, so people’s willingness to share data is itself an interesting research question.

### 2.2. Willingness to Share Data

As detailed as such data can be, participation rates in such experiments show diverse results, but generally, they are rather low when it comes to passive data collection. In what follows, we will refer to the participation rate as “willingness to participate” or WTP, a commonly used abbreviation in this context. We have collected benchmark data from relevant articles studying WTP in various passive data collection scenarios. As Table 1. shows, WTP is mostly below 50%, both for cases where passive data collection is complemented by a survey and for cases where it is not. Although not evident from this summary, the presence of controlled active data collection had a positive effect on participation, only Bricka et al. [25] conducted an experiment comparatively analysing the presence of active data collection.

Most of these studies required the participant to provide location information when filling out a questionnaire, sometimes just a snippet of it. Yet willingness to share this information is particularly low. This result is perplexing considering that most smartphone users share their location data with other apps (often not even in the context of providing location information). Google services, shopping apps are a typical example of location data users.

The only outlier in this table is the study reported by Scherpenzeel [30], where the participation rate is suspiciously high. The participants in this study were panellists who had already completed a larger survey panel for the institution, so there was neither a trust barrier to overcome nor an increased participation burden.

Mulder and de Bruijne [15] went deeper in their study and surveyed behaviour on a 7-point scale (1-very unlikely to participate; 7-very likely to participate) for different data collection types. In their sample, the mean willingness to participate in passive data collection was 2.2, indicating a very low willingness of respondents to participate. In the same study, they found a mean of 4.15 for participating in a traditional PAPI survey study and 3.62 for completing the survey via an app. Thus, the difference between the different ways of completing the survey was not large, but the inclusion of passive data collection had a strong negative impact.

Given the participation rates for regular surveys in general, these even lower numbers are not very surprising. However, to conduct a successful experiment with an acceptable participation rate, it is important to collect the causes that lower the participation rate. In the following, we will look at some factors that have been analysed in different studies.

### 2.3. Importance of Institutional Trust 

Trust in the institution collecting the data was found to be a key factor in the willingness to share data [21,31,32]. Several studies have examined the researching institution’s role on willingness to share passive data. Participants’ main concern regarding data collection is the privacy of their data. It is important to emphasise that a brief indication that the data will not be shared with third parties does not really generate trust among the users of an application, but rather the provider of the application influences it. 

Keusch et al. found that people are about twice as likely to trust research institutions not to share their sensitive data [21]. They measured WTP using an 11-point scale in a survey of panellists. By halving this scale to obtain a dichotomous WTP variable, they found that WTP was similar for all three types of institutions (ranging from 33.1% to 36-9%). However, in their further analysis, they found that WTP was significantly higher in the case of universities and statistical offices than market research firms. Note, however, that in this study no participants downloaded an app, these results only show theoretical readiness. 

Struminskaya et al. found similar results in their study [32], where they tested hypotheses comparing WTP for universities, statistical offices, and market research companies. They found that the WTP reported by respondents is significantly highest for universities, followed by statistical offices and finally for market research firms. 

A practical result for this factor can be found in the study by Kreuter et al. [18], where participants were asked to download a research app sponsored by a research institution on their phone. In their study, they found a WTP of 15.9%, which was the app download rate.

### 2.4. Control over the Data Connection

Passive data collection poses some risk to the user due to the lack of control over the data collection. Here, we consider the ability to temporarily suspend passive data collection from the app as “control” over data collection. Of course, it is possible to prevent an app from collecting data (disabling location services or turning off the mobile device’s Wi-Fi antenna), but here we refer to the case where the experimental app provides in-built ability by design to suspend data collection. 

The best recent example of such an application was provided by Haas et al., where subjects could individually choose which data the application should collect [33]. In their application, users had to give their consent to individual properties that the application could record: Network quality and location information; interaction history; social network properties; activity data, smartphone usage (Figure 13.2/a in [33]). They found that only a very small percentage (20%) of participants changed these options after installing the app and only 7% disabled a data sharing feature.

In their study, Keusch et al. specifically asked about WTP when the participant was able to temporarily suspend the data collection component of the app altogether and found a positive correlation [21]. This differs from the total control that Haas et al. (2020) offered participants, as they allowed the “level” of sharing to be adjusted rather than turned on and off. [33]

Another form of control, the ability to review and change recorded data, was present in Struminskaya et al.’s survey. The corresponding indication in their survey was rather vague, and probably that is why they did not find a significant effect for it [32].

### 2.5. Incentives

Another way to improve WTP is to provide monetary incentives for participation. Haas et al. focused their analysis on different types of incentives paid at different points in an experiment [33]. Incentives can be given in different time frames for different activities of the participants. In terms of time frame, it is common to offer incentives for installing a research application, at the end of the survey, but it is also possible to offer recurring incentives. Another option for incentives is to offer them based on “tiers” of how much data participants are willing to provide. 

In their study, Haas et al. also examined the impact of incentives on the installation and rewarded sharing of various sensor data. There was a positive effect of initial incentives, but interestingly, they did not find the expected positive effect of incentives on granting access to more data-sharing functions. Another interesting finding was that a higher overall incentive did not increase participants’ willingness to have the application installed over a longer experimental period. 

In addition to these findings, their overall conclusion was that the effects of incentives improve participation similar to regular survey studies. The results of Keusch et al. also support this finding [21].

### 2.6. Other Factors

Keusch et al. [21] found that a shorter experimental period (one month as opposed to six months) and monetary incentives increased willingness to participate in a study. As another incentive, Struminskaya et al. [32] found that actual interest in the research topic (participants can receive feedback on research findings) is also a positive factor for increased level of participation. 

Finally, participants’ limited ability to use devices was also found to be a factor in the study by Wenz et al. They found that individuals who rated their own usage abilities as below average (below 3 on a 5-point scale) showed a significantly lower willingness to participate, especially in passive data collection tasks. [23] On the other hand, those who reported advanced phone use skills were much more willing to participate in such tasks. 

Although not necessarily related to age, Mulder and Bruinje found in another study that willingness to participate decreased dramatically after age 50 [15]. These results indicate that usability is important when designing a research application. 

As these results show, there are many details to analyse when designing an experiment that relies on passive data collection. Some of the studies used surveys to uncover various latent characteristics that influence willingness to participate, while others conducted a working research application to share practical usage information.

Given that many studies reported low WTP scores, we concluded that it is very important to conduct a preliminary study before elaborating the final design of such an experiment. Therefore, the goal of this work is to figure out how we can implement a research tool that motivates participation in the study and still collect a useful amount of information.

## 3. Methods and Design

To collect the information on WTP needed to design and fine-tune our research ecosystem and its user interface components, we decided to conduct an online vignette survey using a representative sample of smartphone users in Hungary. In this section, we first formulate our research hypotheses and then present our methods and models for the hypotheses.

### 3.1. Research Questions

Because the focus of our study is exploratory in nature, we did not formulate explicit research hypotheses, but designed our models and the survey to be able to answer the following questions:Q1.What is the general level of WTP in a passive data collection study?In order to have a single benchmark data and provide comparison with similar studies we asked a simple question whether respondents would be willing to participate or not in a study that is built on smartphone based passive data collection.Q2.What features of the research design would motivate people to participate in the study?We included several questions in our survey that address key features of the study: the type of institute conducting the experiment, type of data collected, length of the study, monetary incentives, and control over data collection. We wanted to know which of these features should be emphasized to maximize WTP.Q3.What kind of demographic attributes influence WTP?As mentioned in previous studies, age may be an important factor for participation in our study, but we also considered other characteristics, such as gender, education, type of settlement, and geographic region of residence.Q4.What is the role of trust-, skills-, and privacy related contextual factors on WTP?As previous results suggest, trust, previous (negative) experiences and privacy concerns might be key issues in how people react to various data collection techniques. We used composite indicators to measure the effect of interpersonal and institutional trust, smartphone skills and usage, and general concerns over active and passive data collection methods on WTP.

### 3.2. Survey and Sample Details

Data collection for this study was conducted by a market research firm using its online access panel of 150,000 registered users. The sample is representative of Hungarian Internet users in terms of gender, age, education level, type of settlement and geographical region. The online data collection ran from 9 to 20 June 2021. The average time to complete the survey was 15 min. Basic descriptive statistics of the sample are shown in Table A1 in Appendix A. 

Apart from a few single items, the survey consisted of thematic blocks of multiple-choice or Likert-scale questions. Among others, we asked respondents about interpersonal and institutional trust, general smartphone use habits, and concerns about various active and passive digital data collection techniques using smartphones. The items on trust in the survey were adapted from the European Social Survey (ESS), so they are well-tested and have been used for a long time. With the exception of the last block of the questionnaire, all questions were the same for all respondents. In the last block, a special factorial survey technique was used to ask questions about willingness to participate in a hypothetical smartphone based passive data collection study [34,35].

The factorial survey included situations, called “vignettes”, in which several dimensions of the situation are varied. The vignettes described situations of a hypothetical data collection study and respondents had to decide how likely they would be willing to participate. An example of a vignette is shown in Box 1, with the varying dimensions of the situations underlined, and the exact wording of the outcome variable (WTP).

Box 1An example of a vignette.Imagine the Research Centre for Social Sciences, Hungarian Academy of Sciences Centre of Excellence invites you to participate in research where you are asked to install an application on your smartphone. The research is designed to help researchers better understand the spatial movement of people. The research will last for one month. Upon completion of the research, participants will be given vouchers worth HUF 5000. At any time during the research, you will have the option to turn off or temporarily suspend the application, and view the data collected from your device and authorise its transfer to our servers.Based on the characteristics just presented above, how likely do you think you are to participate in the research and download the app to your phone? Please mark your answer on a scale of 0 to 10, where 0 means you would certainly not participate and 10 means you would certainly participate in the research.


**Certainly would not participate**

**Certainly would participate**
012345678910

Six dimensions were varied in the vignettes with the following values:The organizer of the research: (1) decision-makers, (2) a private company, (3) scientific research institute.Data collected: (1) spatial movement, (2) mobile usage, (3) communication habits, (4) spatial movement & mobile usage, (5) spatial movement & communication habits, (6) mobile usage & communication habits, (7) all three.Length of the research: (1) one month, (2) six months.Incentive: (1) HUF 5000 after installing the application, (2) HUF 5000 after the completion of the study, (3) HUF 5000 after installing the application and HUF 5000 after the completion of the study.Interruption and control: (1) user cannot interrupt the data collection, (2) user can temporarily interrupt the data collection, (3) user can temporarily interrupt the data collection and review the data and authorize its transfer.

Following Jasso, the creation of the vignettes proceeded as follows [35]: First, we created a “universe” in which all combinations of the dimensions described above were present, which included 378 different situations. From these 378 situations, we randomly selected 150 and assigned them, also randomly, to 15 different vignette blocks, which we call decks. Here, each deck included 10 different vignettes and an additional control vignette to test the internal validity of the experiment. The content of this last vignette was the same as a randomly selected vignette from the first nine items previously evaluated. The results show a high degree (64%) of consistency between responses to the same two vignettes, suggesting a satisfactory level of internal validity (see Appendix B for details on the analysis of this test). In this manner, each respondent completed one randomly assigned deck with 10 + 1 vignettes. In total, 11,000 vignettes were answered by 1000 participants. (Data from the 11th vignette were excluded from the analysis). The descriptive statistics of the vignette dimensions are presented in Table 2.

This technique allowed us to combine the advantages of large surveys with the advantages of experiments. Due to the large sample size, the analysis has strong statistical power, and we can also dissociate the effects of different stimuli (dimensions) using multilevel analysis. [36,37] Thus, we can examine the effect of multiple variables on the outcome variable (WTP measured on a 0 to 10 scale). 

In addition to vignette-level variables, we also included respondent level variables in the model to examine how individual characteristics influence the effects of vignette dimensions on participation. We included both respondent-level sociodemographic variables and attitudinal variables in the model. The sociodemographic variables were gender (coded as males and females); age; education with four categories (primary school or lower, vocational, high school, college); place of residence with the type of settlement (capital city, county seat, town, village); and the seven major regions of Hungary (Central Hungary, Northern Hungary, Northern Great Plain, Southern Great Plain, Southern Transdanubia, Central Transdanubia, and Western Transdanubia).

The attitudinal variables we used in the models were the following: For how many types of activities does the respondent use their smartphone. We queried 15 different activities (see Table A2 in Appendix A for the full list of activities) and simply counted the activities for which the respondent actively uses his or her smartphone.

The personal trust variable shows the average of responses for three trust-related items (see Table A3 in Appendix A for details) measured on a scale from 0 to 10, where 0 represents complete distrust and 10 represents complete trust. We performed the same calculation for trust in institutions. We listed several institutions (see Table A4 in Appendix A for the full list) and asked respondents to indicate their level of trust on a scale of 0 to 10, where “0” means they do not trust the institution at all and “10” means they trust it completely. 

We also included several digital data collections techniques and asked respondents how concerned they would be about sharing such information for scientific research, emphasizing that their data would only be used anonymously and in aggregated format without storing their personal information. The response options were 1 to 4, with 1 meaning “would be completely concerned” and 4 meaning “would not be concerned at all.” In total, we asked about 18 different active and passive data collections (see Table A5 in Appendix A for the full list of items), from which we formed two separate indices: 6 items measured active, and another 12 items measured passive data collection techniques. For both composite indicators, we counted scores of 1 and 2 (i.e., those more likely to indicate concern). For the statistical proof of the indices’ internal consistency, we performed Cronbach’s alpha tests, which proved to be acceptable in each case. 

In addition to the sociodemographic variables and the composite indices, we added two other variables: the time respondents spend online and use their smartphones (in minutes) on an average day.

### 3.3. Analysis and Models

The analysis could be divided into four parts. First, we simply checked for the descriptive results of the benchmark variable showing the general level of willingness to participate in a smartphone-based passive data collection study.

In the next step we constructed variance component models, to understand the direct effect of the decks by calculating the total variance in the vignette outcome that is explained by respondent characteristics vs. deck of vignettes. 

In the second part of the analysis, we created three linear regression models. These models were multilevel models because the analyses were conducted at the vignette level, but each set of 10 vignettes was completed by the same subject. Thus, the assumption about observational independence—which is required in the case of general linear regression—was not made. To control for these dependencies, we used multilevel mixed models. In the first model, we included only the independent variables at the vignette level. Then, in a second step, we added respondents’ sociodemographic characteristics, as we assumed that these influence respondents’ willingness to participate. In a third step, we additionally included composite indices of the attitudinal variables at the respondent level. In the final step of the analysis, we added cross-level interaction terms to the model to examine how vignette-level dimensions are varied by respondent-level characteristics.

## 4. Results and Discussion

In general, 50 percent of respondents would participate in a study that includes the passive collection and sharing of data from the respondents’ smartphones. The online access panel that was used for this survey includes panellists who from time to time are taking part in active data collection (i.e., filling out online surveys through their PCs, laptops or smartphones), so they presumably comprise a rather active and more motivated segment of the Hungarian internet users. But one in two of them seems to be open for passive data collection as well. (Q1) 

The vignettes used in the survey was designed to understand the internal motives and factors behind that shape the level of willingness. In the first step of this analysis, we built two only intercept models, in which the dependent variable was the outcome and there were no independent variables, but the control for the level of decks and the level of respondents. Based on the estimates of covariance parameters, we could conclude on the ratio of explained variance by the different levels. The variance component models revealed that 77.6 percent of the total variance in the vignette outcome is explained by respondent characteristics and 1.4 percent is explained by the deck of vignettes. Thus, the effect of the decks (the design of the vignette study) is quite small.

We then created three multilevel regression models. (Table 3). In the first one (Model 1), we included only the independent variables at the vignette level. Then, in a second step (Model 2), we added the socio-demographic characteristics of the respondents, as we assumed that they influence the respondents’ willingness to participate. In a third step (Model 3), we also included composite indices of respondent-level attitudinal variables. Table 3 shows the results of the three models.

Results of Model 1 revealed that compared to policymakers, respondents are significantly more likely to participate in research conducted by a private company (with an average of 0.15 points on a scale of 1 to 10) or a scientific research institute (with an average of 0.29 points. People are more willing (with an average of 0.68 points) to participate in a study that lasts only one month-compared to one that lasts six months. And not surprisingly, they would be more likely to participate in a study if they were paid twice instead of once-by about 0.44 points. The chance of participating is highest if the user can suspend the data collection at any time and view the collected data when needed. The two options of no suspension and suspension but no control over the data showed a lower chance of participation (with an average of 0.59 and 0.13 respectively). Interestingly, there were no significant differences between the purpose of the data collection and thus the type of data collected. Compared to the reference category, where all three types of data are requested, none of the other single types of data collection showed significantly lower or higher level of participation (Q2).

We included respondents’ sociodemographic variables in Model 2. Including respondents’ sociodemographic variables did not really change the effect of the vignette dimensions. Interestingly, none of the sociodemographic characteristics have a significant effect on participation, with the exception of age: in accordance with previous research, older individuals are less likely to participate. The probability of participation decreases by 0.04 points with each additional year (Q3).

In Model 3, we added respondents’ attitudinal indices to the model. The addition of the respondent-level attitudinal variables did not really change the effects of the variables compared to the previous models. Of the attitudinal variables, none of the trust indices appear to have a significant effect, however smartphone use and concerns about passive data collection do change the likelihood of participation. This is because the more activity and the longer time someone uses a smartphone, the more likely they are to participate in such a study. The more types of passive data collection someone has concerns for, the less likely they are to participate (Q4).

### 4.1. Varying Effects of the Vignette Level Variables among Respondents

In the next step we tested the vignette-level variables that had a significant effect on willingness to participate to see if their effect differed across respondents. These variables were the length of the study, the organizer of the research, the type of incentive, and the possibility of suspension and control. We set the slope of these variables to random (one at a time, separately in different models) and tested whether they were significant, that is, whether the effects varied across respondents. To achieve convergent models, we transformed some of the vignette-level variables into dummies. The transformation was based on the results of the previous models and categorized together those values that showed the same direction of effects. The organizer of the research was coded as: (a) private company or scientific research institute vs. (b) policymakers. The type of incentive was categorized as (a) only one vs. (b) two incentives given. The opportunity of suspension was transformed into (a) no opportunity or there is opportunity vs. (b) opportunity with control over transferred data. The results showed that all of these variables had significant random slopes, so all effects varied between respondents.

### 4.2. Interaction of Vignette- and Respondent Level Variables on the Willingness to Participate

We also tested all interaction terms of those vignette variables that had significant random slopes (length of the study, organizer of the research, type of incentive, and possibility of suspension and control) with those respondent level variables that had a significant effect on willingness to participate (age, smartphone use, and concerns about the passive nature of data collection).

Of the twelve interactions tested, six proved to be significant. Figure 1 shows the nature of these interactions with the means of the predicted values. For illustration purposes, we divided each ratio-level variable into two categories and used their mean as cut values.

We can observe that shorter research duration predicts higher probability of participation, but this effect is stronger for those who use their smartphone for more types-compared to those who use it for fewer. (a) Interestingly, the effect of the length of study is stronger among those with fewer concerns about passive types of data collection and weaker among those with more concerns. (b) When we consider the number of incentives, we can see that while two incentives generally increase the odds of participation compared to only one incentive, this effect is stronger among those who use their smartphone for fewer types of activities and weaker among those who use it for more activities. (c) In addition, the effect of the incentive is stronger among those who have fewer concerns about passive data collection and weaker among those who have more concerns. (d) In the original model (Model 3, Table 3), we could see that someone is more likely to participate if they can interrupt the study when they want to and if they can review the data collected about them-compared to simply suspend or even not being able to suspend. Based on the interactions, this effect is stronger for younger individuals than for older individuals. (e) When we account for smartphone usage, we see that the effect of type of suspension disappears for those with lower smartphone usage and persists only for those who use their smartphone for more tasks (f).

## 5. Conclusions

With this study, we aimed to continue the series of analyses examining users’ attitudes toward passive sensors- and other device information-based smartphone data collection.

Overall, our results are consistent with findings of previous research: We found evidence that a more trusted survey organiser/client, shorter duration of data collection, multiple incentives, and control over data collection can significantly influence willingness to participate. The results also show that apart from age (as major determinant of digital technology use and attitudes towards digital technologies), demographic characteristics alone do not play an important role. This finding might be biased by the general characteristics of the online panel we used for the survey, but they might come as an important information for future studies that aim for representativeness of the online (smartphone user) population. 

Contrary to our preliminary expectations, trust in people and institutions alone does not seem to have a notable effect. This is especially noteworthy given the fact that the Hungarian society has generally lower level of personal and institutional trust compared to Western and Northern European countries. However, general attitudes toward technology, the complexity and intensity of smartphone use, and general concerns about passive data collection may be critical in determining who is willing to participate in future research. 

Asking questions on future behaviour of people in hypothetical situations have obvious limitations. In our case, this means that there is a good chance that we would get different results if we asked people to download an existing, ready-to-test app and to both actively and passively collect real, personal data from users. We were mainly interested in people’s feelings, fears and expectations that determine their future actions, and we suggest that our results provided valid insights. 

It should also be mentioned that in this research we focused mostly on the dimensions analysed in previous studies and included them in our own analysis. Of course, there are many other important factors that can influence the willingness of users to participate. Our aim was therefore not to provide a complete picture, but to gather important aspects that could enrich our collective knowledge on smartphone based passive data collection and inform our own application development process.

## Figures and Tables

**Figure 1 sensors-21-06085-f001:**
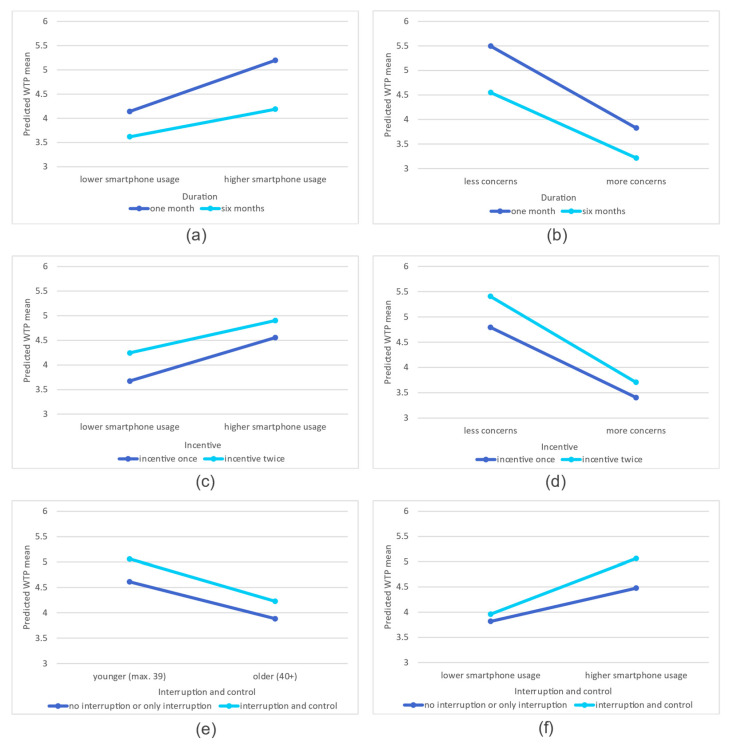
Predicted values of cross-level interactions. (**a**) Length of study with smartphone usage. (**b**) Length of study with concerns overs passive data collection. (**c**) Number of incentives with smartphone usage. (**d**) Number of incentives with concerns over passive data collection. (**e**) Interruption and control with age. (**f**) Interruption and control with smartphone usage.

**Table 1 sensors-21-06085-t001:** The ratio of willingness to share data in selected studies.

Study	Passive Data Collection Contents	Willingness to Participate (WTP)	Note
Biler et al. (2013) [26]	GPS data	8%	
Kretuer et al. (2019) [21]	mobile phone network quality, location, interaction history, characteristics of the social network, activity data, smartphone usage	15.95%	
Toepoel and Lugtig (2014) [27]	GPS data	26%	one-time, after a survey
Bricka et al. (2009) [25]	GPS with survey GPS only	30–73% 12–27%	in this study, the participants would fill multiple surveys
Pinter (2015) [28]	Location	42%	this was only claimed willingness not actual downloads of an application
Revilla et al. (2016) [22]	GPS data	17.01–36.84%	this value is the min-max willingness rate of mobile/tablet users from the following countries: Argentina, Brazil, Chile, Colombia, Spain, Mexico, Portugal
Revilla et al. (2017) [29]	Web activity	30–50%	
Scherpenzeel (2017) [30]	Location (GPS, Wi-Fi, cell)	81%	
Wenz (2019) [23]	GPS Usage	39% 28%	

**Table 2 sensors-21-06085-t002:** The descriptive statistics of the dimensions of vignettes.

**Organizer of the research**						
decision-makers		private company			scientific research institute	
35.3%		34.2%			30.4%	
**Data collected**						
spatial movement	mobile usage	communication habits	spatial movement & mobile usage	spatial movement & communication habits	mobile usage & communication habits	all three
12.3%	15.8%	13.5%	18.2%	11.3%	15.1%	13.8%
**Length of the research**						
one month				six months		
48.1%				51.9%		
**Incentive**						
HUF 5000 after installing the application		HUF 5000 after the completion of the research			HUF 5000 after installing the application and HUF 5000 after the completion of the research	
33.7%		32.6%			33.6%	
**Interruption and control**						
user cannot interrupt the data collection		user can interrupt the data collection			user can interrupt the data collection and has control over their data	
34.2%		31.7%			34.2%	
**Willingness to participate**						
min.		max.		mean	standard deviation	
0		10		4.50	3.65	

Notes: the vignette level data, N = 10,000.

**Table 3 sensors-21-06085-t003:** Multilevel Regression Models on the Willingness to Participate.

h	Dependent Variable: Willingness to Participate		
	Model 1	Model 2	Model 3
Intercept	5.85 *	8.12 *	6.94 *
	(0.13)	(0.82)	(1.04)
Vignette level variables			
Organizer of the research (ref: decision makers)			
Private company	0.15 *	0.15 *	0.22 *
	(0.04)	(0.04)	(0.04)
Scientific research institute	0.29 *	0.29 *	0.36 *
	(0.04)	(0.04)	(0.05)
Data collected (ref: all three)			
Spatial movement	0.09	0.09	0.05
	(0.07)	(0.07)	(0.07)
Mobile usage	−0.11	−0.11	−0.14 *
	(0.06)	(0.06)	(0.07)
Communication habits	−0.02	−0.01	−0.09
	(0.07)	(0.07)	(0.07)
Movement & usage	−0.02	−0.02	−0.06
	(0.06)	(0.06)	(0.07)
Movement & communication	0.03	0.03	−0.05
	(0.07)	(0.07)	(0.08)
Usage & communication	0.00	0.00	−0.02
	(0.06)	(0.06)	(0.07)
Length of the research (Ref: one month)	−0.68 *	−0.68 *	−0.74 *
Incentive (ref: after downloading the app & after the end of the research)	(0.03)	(0.03)	(0.04)
After downloading the app	−0.43 *	−0.43 *	−0.44 *
	(0.04)	(0.04)	(0.05)
After the end of the research	−0.47 *	−0.47 *	−0.49 *
	(0.04)	(0.04)	(0.05)
Interruption and control (Ref: user can interrupt the data collection and has control over their data)			
User can interrupt the data collection	−0.13 *	−0.13 *	−0.17 *
	(0.04)	(0.04)	(0.05)
User cannot interrupt the data collection	−0.59 *	−0.59 *	−0.64 *
	(0.04)	(0.04)	(0.04)
Respondent level socio-demographic variables			
Gender (Ref: men)		−0.30	−0.16
		(0.22)	(0.22)
Age (+: older)		−0.04 *	−0.02 *
		(0.01)	(0.01)
Education (+: higher)		−0.24	−0.20
		(0.13)	(0.14)
Type of settlement (+: smaller)		0.13	−0.02
		(0.12)	(0.12)
Region (Ref: Western Transdanubia)			
Central Hungary		0.07	0.29
		(0.41)	(0.43)
Northern Hungary		0.08	0.52
		(0.47)	(0.48)
Northern Great Plane		0.02	0.46
		(0.45)	(0.48)
Southern Great Plane		0.44	0.62
		(0.46)	(0.48)
Southern Transdanubia		0.28	0.35
		(0.48)	(0.52)
Central Transdanubia		0.72	0.84
		(0.47)	(0.50)
Respondent level attitude indices			
Smartphone activities (+: multiple)			0.10 *
			(0.04)
Personal trust (+: high)			0.07
			(0.05)
Institutional trust (+: high)			0.12
			(0.07)
Time spent online on an average day (minutes)			0.00
			(0.00)
Time spent using their smart phone on an average day (minutes)			0.00
			(0.00)
Number of active data collection mentioned as rather worrying			−0.06
			(0.09)
Number of passive data collection mentioned as rather worrying			−0.20 *
			(0.04)
AIC	44,997.2	44,977.2	37,066.5
BIC	45,011.8	44,991.8	37,080.7
Observations	100,000	10,000	10,000

Notes: Standard errors in parentheses. * *p* < 0.001.

## Data Availability

The data presented in this study are available on request from the corresponding author. The data are not publicly available due to contractual restrictions with the fieldwork agency conducting the online survey.

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
