# Peer review of "Attitudes towards Participation in a Passive Data Collection Experiment"

_sensors, 2021, doi:10.3390/s21186085_

Round 1
Reviewer 1 Report
In this paper, the authors conducted an interesting survey to explore people’s attitudes and willingness to participate in passive data collection. This is a fundamental research question and needs to be investigated to facilitate people’s participation in future data collection. I think the paper has the potential to be published. However, this manuscript can still be improved from the following perspectives.
- The authors are suggested adding a table to document the descriptive statistics of their survey respondents.
- The process of variance component modeling is too sketchy. Please add descriptions and details to explain why and how you perform variance component modeling in this study.
- The result of the survey is not documented. The authors are suggested adding a table to highlight the summarized survey results.
- Lines 378-380 “which revealed that 77.6 percent 378 of the total variance in the vignette outcome is explained by respondent characteristics 379 and 1.4 percent is explained by the deck of vignettes.” It should be part of your result and documented in section 4.
- Some coefficients (e.g., Organizer of the research) are missing from your Table 1.
- Based on table 2, it seems that you only considered gender, age, education, type of settlement, and the region as the socio-demographic variables. However, typically, we will include more variables (e.g., income, occupation) to characterize the survey respondents. Have you examined these variables? If not, please discuss the potential influence of these variables on your modeling result.
- Figure 1 is too blur, and the label is missing for the y-axis.
- In Line 470, “Figure 2 shows the nature of these interactions with the means of the predicted values.”, However, I didn’t find Figure 2 from this manuscript.
- The authors designed three options for the incentives, including HUF 500 after installation, HUF 5000 after the completion of research, and two HUF 5000 after both installation and research. Please explain why you fixed the amount to HUF 5000. And it seems to be an obvious answer that people like to be paid twice, leading to most respondents would choose the third option. And it would be more interesting to see how people’s attitude changes when applying different incentives.
- The authors are suggested discussing the limitation of the survey and how to improve it in future works.
Reviewer 2 Report
The papers presents an exploratory study conducted in Hungary using factorial design based online survey to explore the willingness to participate in a future research project based on active and passive data collection via smartphones.
The authors conducts an active data collection and show the results. A passive and complementar data collect via smartphone should be conducted to be a study more complete and more attached to this journal. Results about these data should be discussed too.
Related work must be papers more recent.
The authors must answer all their research questions. The results do not help this aspect. For example, the question 1 "What is the general level of WTP in a passive data collection study?" is not answered. This must be clear in the paper.
Round 2
Reviewer 1 Report
The authors have addressed all my comments.
Author Response
Dear Reviewer,
once again, thank you for your earlier comments and suggestions. We are glad that you were satisfied with the changes and replies we submitted in the first review round.
Reviewer 2 Report
The authors affirm that "tried to cite papers that have meaningful results in examining various factors that might have an influence on people’s willingness to share their data passively collected by their smartphones. We refer to several articles that were published recently (in the past 2-3 years)."
Table 1 lists old references (just two recent 2019). Therefore, I suggest the authors provide the systematic review protocol, including search string, inclusion and exclusion criteria, etc.
Author Response
Dear Reviewer,
Table 1 was intended as a narrow selection of sources in which the level of willingness to participate was investigated either directly (people downloading an app) or indirectly (through surveys) in passive data collections. We limited the scope of this table to those articles that provided an explicit number (or range).
We have added an additional sentence to the text (line 126) to make this clearer.
Of course, the number of articles we refer to in the text is much larger and includes some recent work.